# Analyzing an Appropriate Blend of Face-to-Face, Offline and Online Learning Approaches for the In-Service Vocational Teacher’s Training Program

**DOI:** 10.3390/ijerph191710668

**Published:** 2022-08-26

**Authors:** Muhammad Zaheer Asghar, Muhammad Naeem Afzaal, Javed Iqbal, Hafiza Amina Sadia

**Affiliations:** 1Crafts Sciences Unit, Department of Teacher Education, University of Helsinki, 00100 Helsinki, Finland; 2Department of Education, University of Management and Technology, Lahore 54770, Pakistan; 3Department of Law, Bahria University, Islamabad 04403, Pakistan

**Keywords:** blended learning, online learning, offline learning, face-to-face learning, in-service vocational teachers

## Abstract

Blended learning approaches are considered as the most viable for the delivery of training to remote areas and accessing learners at a mass level. Blended learning is a combination of different learning approaches to facilitate the learners’ needs. The National Vocational and Technical Training Commission (NAVTTC) conducted an in-service vocational teachers’ training program through blended learning approaches in Pakistan. This study aimed to find the appropriate blend of face-to-face, online, and offline learning approaches for the training of in-service teachers in Pakistan. A mixed methods research approach was adopted. A survey collected data from 781 in-service vocational teachers who participated in training programs through blended learning approaches. The ANOVA test was applied to find the difference of the training participants’ attitude toward different modes of learning. It was found that trainees had more positive attitude toward a face-to-face learning approach than online and the online learning approach than offline learning. Semi-structured interviews were also conducted with training participants, admission and placement officers, and principals. They also endorsed that face-to-face learning approaches must be given more weight than online, and the online approach should have more weight than the offline approach. This study has practical implications for technical education and vocational training (TVET) institutes in developing countries such as Pakistan to design blended learning approaches for the training of in-service vocational teachers. Future research may be conducted on the effectiveness of in-service vocational education teachers through blended learning.

## 1. Introduction

The role of skill development among the youth is very important in developing a stable society [1]. In order to provide skills to the youth, the role of teachers in the technical and vocational sector is very important and requires their development in the pedagogical aspect, which further improves classroom teaching [2]. Schunk et al. [3] viewed pedagogical training as an effort toward the improvement in the educational system. The National Vocational and Technical Training Commission (NAVTTC), in coordination with GIZ and training providers such as the Punjab Vocational Training Council (PVTC) and Technical Education and Vocational Training Authorities (TEVTAs), has launched a professional development program of in-service teachers in the field of pedagogy under the National Skills Strategy (NSS, 2009) [1]. The said program adopted an innovative approach of blended learning to train teachers to improve their teaching skills across Pakistan. Professional development programs have a common theme of technology integration [4], which if implemented only to provide technology usage related training to teachers, it will work for technology integration in the classroom [5]. While switching to blended learning practices, most professional development programs have focused on the technology usage rather than connecting it to pedagogical practices [6]. The teacher training programs that have focused on pedagogy and content knowledge, along with the technology usage training, proved to be more successful in comprehending blended learning compared to the professional development courses that merely focused on the technology usage skills [7]. Different studies [6,8,9,10] have shown that engaging teachers in blended learning programs create empathy in teachers for their students. Training programs for teachers in which both technology and pedagogical contents are focused promote long lasting effects on technology application in the classroom [5].

The blended learning approach has been used to enhance the pedagogical skills of vocational teachers in Pakistan. It was suggested to integrate blended learning training programs at a slower pace, which will produce benefits such as an increase in meaningful data collection, facilitate quick learning as well as greater engagement and involvement of students in personalized learning. Vaughan et al. [11] explained that teachers who are involved in blended learning classes should pay great attention to the teaching methods and teaching learning process. When viewed through the lens of the outcome-based educational paradigm, blended learning has many advantages such as a pedagogical strategy and a syncretic form of education delivery [12]. Blended learning programs focus on a learner centered approach that provides learners with independence in their study and pace of learning [13,14]. According to Dangwal [15], communication skills, responsibility, and motivation for learning have been observed by researchers in the students who were attending classes with a blended learning approach. Blended learning provides teachers and students with a great deal of flexibility, which is a huge benefit [13,14]. Students in higher and secondary education evaluated flexibility as a major benefit of this new method since it gave them the freedom to choose when, where, and how they performed and completed their assigned tasks and responsibilities [14,16]. Borup and Stevens [17] also endorsed its flexibility as a great benefit of blended learning, choosing educational content, delivery approach, and scheduling as positive outcomes of blended learning. Trainees from far-flung areas received teaching support with the help of the blended learning technique [8,15]. Trainees who were unable to attend traditional educational institutions regularly benefited from this approach as it reached them at their doorstep [15]. According to Dangwal [15], if learners are unable to reach the educational institution, then the institution should reach them. Students at risk (Pytash and Zenkov [18] and Repetto, et al. [19]), with severe health issues (Black and Thompson [20]), and students in long-term incarceration have achieved good results due to the blended learning technique. According to Black and Thompson [20], students who cannot maintain regular traditional schools benefit from the principle of the flexibility of the blended learning program. According to Repetto, Spitler, and Cox [19], research has shown that the drop out ratio of students was reduced with a blended learning approach compared to the traditional schools, as blended learning provides the flexibility of online and face-to-face classes. Therefore, the blended learning approach has enormous benefits that cannot be denied, but at the same time poses certain challenges.

Previous research has studied blended learning in the general education sector and measured the e-learning content, student achievement, and application of the innovative approach. The majority of research on blended learning approaches have been undertaken in higher education, while research on blended learning approaches in the vocational sector is limited [21,22,23]. The use of ICT resources is a major factor in the blended learning approach being the need of the hour for managing online and offline activities. There are several researchers who are of the opinion that the effectiveness of blended learning is limited because of the challenges associated with the students and teachers [13,20,24,25]. The main challenge is the time needed by teachers to grasp the technology related to the blended learning approach. The other challenge is the sudden shift in teaching practices from traditional to blended learning [26]. It is also rare for there to be an in-service teacher training program offering a blended learning approach [14]. The design and delivery techniques are also very poor, posing a great challenge to this innovative approach [14]. When students went back to their home areas, they found very little Internet and technology facilities [27]. Vaughan, Reali, Stenbom, Van Vuuren, and MacDonald [11], being researchers who have gone through the comprehensive research on blended learning, identified that many of the challenges directly or indirectly affecting the results of blended learning were the limited teacher training, the lack of skills in using innovative technology, the financial burden, school vision, and challenge for teachers to shift from the role of content provider to facilitator.

Researchers are of the opinion that the great challenges in receiving the full benefits of blended learning are the technical requirements, the method of conducting training, knowledge about the theory of blended learning, and its implementation [20,24,27,28]. Rice and Skelcher [29] argued that in-adequate education policy and tremendous technology advancement were real challenges in the implementation of a blended learning approach. This is because of the fact that information technology is changing daily, which poses a great challenge for policy makers to cope with the quick adoption of technology and the implementation of the blended learning technique. The authors in [30] were of the opinion that the implementation of an innovative blended learning approach forced the policy makers to counter the issue of student attendance who were away from the institutes, and similarly design a framework of standards for the blended learning approach and ensure the availability of the required technical resources. Instructors are urgently required to bear certain points in mind while facing major challenges in formulating or designing an effective system of blended learning [31]. The interaction among the trainees and teachers, the flexibility of time and place for students, ensuring an effective learning processes, and providing a motivational and effective climate that is useful for learning are the major challenges in implementing this innovative approach [31].

This research aimed to find the combination of different approaches (i.e., online, offline, and face-to-face) for organizing in-service vocational teacher training programs through blended learning in a developing country such as Pakistan. This research is significant in three ways: first, it measures the combination of different blended learning components according to the need of in-service vocational teacher training program; second, it presents a robust statistical analysis of a big dataset to generalize the results; and third, it has wider implications for TVET institutions, skills development authorities, and future researchers in designing instructional strategies for the effective delivery of in-service vocational teacher training through blended learning approaches. The major research question under study is, “what is the appropriate blend of face-to-face, offline, and online learning approaches for an in-service vocational teacher training program?”.

The introduction section of the study is followed by the literature review and research methods. The research methods discuss the research approach, questionnaire development, population sample, and data analysis procedures. This is followed by the discussion. Finally, we present the conclusions, limitations, and recommendations.

## 2. Literature Review

Blended learning has been defined in a variety of ways, and its definition has evolved through time, with little debate on the terms online and face-to-face learning [32]. When technology was not incorporated in education, blended learning was traditionally thought to be the use of multiple teaching techniques in the classroom to aid learning [33]. When face-to-face education is integrated with technology such as e-learning and online learning, students learn more successfully. Nortvig, Petersen, and Balle [32] stated that the focus of blended learning was subsequently switched to the utilization of a mixed learning environment, which included both face-to-face and online learning [33]. The most widely accepted definition is combining online, face-to-face, and offline learning in the same training course [34]. According to Gurley [34] and Vaughan, Reali, Stenbom, Van Vuuren, and MacDonald [11], the teachers often do not understand the definition of the term blended learning and remained confused while defining the term. According to Riel et al. [35], the misinterpretation over the definition of the term blended learning should be considered for all periods of practicing a blended learning pedagogy differently. It was argued and defined that blended learning was the combination of face-to-face and online learnings that included a minimum 30% and maximum 79% of teaching content and activities online [34]. In contrast to the previous definition, the term blended learning should cover at least a 50% training part in face-to-face teaching [32]. Horn and Staker [36] proposed a new meaning to blended learning and argued refraining from time bounding during educational practices online and face-to-face. In their opinion, the term blended learning means that the trainees should learn any time, in part under supervision in a brick-and-mortar location, which means away from home, and in part with an online delivery. In this way, trainees gain some independence and control over the time and place [36]. Alternatively, training courses are considered as online when the training materials are 80% delivered online [34]. The difference of opinions regarding the definition of blended learning mainly revolves around the time and ratio of online and face-to-face learning [37]. According to Dziuban et al. [38] there is ambiguity about the standard definition of blended learning, which also limits its effectiveness in professional development and adoptability in educational practices. Spring and Graham [37] suggested viewing blended learning from a broader perspective for more dissimilarities and individualization of instructions. According to Halverson, Spring, Huyett, Henrie, and Graham [10] teachers should place emphasis on the pedagogical characteristics of blended learning in spite of arguing for face-to-face and online learning features that might support the better educational practices of teachers.

The more important element of the definition of blended learning is the student’s control over the place, time, and path of learning [13,30]. According to Horn and Staker [30], the idea of blended learning is different from the traditional teaching method and provides students with independence to manage their own learning by having control over time for online learning, maintaining self-pacing, and selecting the place and path to learn different concepts online. In light of Parks, Oliver, and Carson [6], the discussion regarding the definition of blended learning should focus on the student’s individuality. It is generally believed by many teachers that only the inclusion of technology in classroom teaching, which is face-to-face, is blended learning [11]. It is mostly confused that blended learning courses are planned one-to-one, which is not true [30]. The use of technology to access online material and different websites in one-to-one planned programs for facilitating learning does not permit the students’ independence and control over time and place [30]. Similarly, scheduling a one-to-one program is not synonymous with blended learning [11,25].

The manner in which learners receive instructions differs amongst blended learning approaches [36]. Blended learning models have different instructional influences, which include the delivery method, role of teachers, space, and schedules [36]. According to Prasad [1], there are four core blended learning models: rotational model, flex model, virtually enriched model, and a la carte model.

### 2.1. Rotation Model

In this strategy, instructors create a schedule that rotates students through different learning techniques such as group work, independent study, and Internet work [30]. Multiple learning modules are designed in a course and students have to select one online module [39]. Learners must stick to a rigid schedule that alternates between different training methods such as classroom instruction, e-learning, and even collaborative group activities and debates [40].

### 2.2. Flex Model

The digital technology for online learning and teacher supervision is the backbone of the flex model [41]. The model enables teachers to change their students’ learning styles on their own as well as expand the instructional resources on a flexible basis [39]. It also contains some group instruction such as group activities, group projects, or individual tutoring led by the instructor [40]. Students accomplish their assignments by following the course materials. Students can contact their teacher online if they have any questions or issues regarding the program’s materials and assigned tasks [42]. The students work at their own pace to complete the work, while the teacher is physically present to oversee the students’ progress [43]. As a result, the current approach is incredibly adaptable, allowing students to move between online and offline learning activities to complete their assignments with the help and advice of their trainers [44]. The strategy is extremely beneficial for the trainees since it gives them ownership of their learning as well as control over their progress [45].

### 2.3. A La Carte Model

This refers to students who, in addition to the traditional course, attend an online course on their own time. The students’ course is conducted outside of the typical classroom setting [36]. In this paradigm, the school’s curricular resources are usually insufficient to meet the students’ learning needs, necessitating the acquisition of one or more courses or disciplines online, which would be fully online outside the classroom [39]. The majority of the instruction is provided by instructor-led training, which is supplemented by online resources [40].

### 2.4. Enriched Virtual Model

Students are occasionally needed to be physically present in the classroom with their professors, although the notion mostly centers on information delivery via the Internet [36]. Students who enrolled in online programs for the a la carte model and enriched virtual models are comparable models [44]. Teachers deliver the instructions and course content online in both approaches [44]. Aside from the similarities, there is a difference: the a la carte model allows students to select one online learning course, but the virtually enriched blend (i.e., online) allows students to meet face-to-face in schools on occasion [40,43].

In-service training of vocational teachers in Punjab adopted the enriched virtual model of blended learning approaches in which trainees were provided with the opportunity of self-paced learning. They were provided with the training materials and tasks, which were monitored by the trainer. The participants had to perform their tasks in face-to-face and self-study mode. The trainer provided feedback on their assignments mostly in online mode and sometimes in face-to-face mode. Group activities and individual tasks were also conducted in face-to-face mode. The assigned tasks were sent to the trainer through online mode by using emails.

### 2.5. Conceptual Framework

We followed the widely accepted definition of blended learning, which is a mixture of one-to-one technology and face-to-face teaching of contents and skills [46]. According to [46,47], the term blended learning means a mixture of face-to-face teaching and online learning. The study in [48] defined blended learning as a combination of online and face-to-face educational practices. They further elaborated that the contextual realities should be considered while defining the term blended learning. Simiyu and Macharia [49] were of the opinion that teachers in the degree program at Moi University should be included by utilizing a blended learning approach and in their viewpoint, blended learning was a mixture of face-to-face teaching and online learning through communication tools such as emails and discussion forums.

#### 2.5.1. Face-to-Face Learning

Classroom teaching is referred to as face-to-face instruction in a real classroom environment [50]. According to Ren et al. [51], course instruction is primarily centered on instructor classroom lectures in the traditional face-to-face teaching mode, and students can only use the remaining time in the classroom and perform practical and group activities. Evaluation is objective in the case of face-to-face learning. It is normally conducted in a systematic manner under the supervision of teachers. In this way, it affects the thinking of the students and blocks their potential abilities. According to Baker and Unni [52], the preference of students from the Asian region for face-to-face learning is higher compared to online and offline learning. Participants feel comfortable in face-to-face sessions as it provides a quick solution to their problems and hands-on experience. Tratnik et al. [53] argued that student satisfaction with face-to-face learning was greater than online learning. In contrast to the current results, research has also shown that students were more satisfied from online learning compared to face-to-face learning [54]. Similarly, [55] discussed the organizational model of online learning along with traditional face-to-face learning.

#### 2.5.2. Online Learning

Web-based learning progress, commonly referred to as online learning, is intended to offer physical classroom-based instructional content over the Internet [56]. The most difficult type of learning in terms of challenges is online, followed by distance learning, and finally, blended learning [57]. According to Bora [58], learning that takes place over the Internet or through electronic technology to obtain instructional content is referred to as online learning. Students can learn and connect with instructors and other students from anywhere in an online environment [59]. Online learning has risen in popularity as a result of its capacity to provide more flexible access to content and teaching at any time and from any location [60]. It uses learning resources such as PowerPoint presentations, lectures, documents, pictures, and videos, etc. to create an independent learning environment for the students [51]. Due to the technological improvements, teaching and learning can now take place in multiple locations online at the same time [61]. Online learning provides solutions to the problems faced in conventional education; however, it is also imposing some problems (e.g., separation, remoteness, adequate feedback, distancing from group members, and lack of seriousness) [62]. A research study showed that 89% of students believed that practical training through an online mode of delivery was not recommendable [63]. In this critical situation, blended learning has emerged as a solution to the problems faced in both approaches as it covers online and face-to-face learning [64]. Training courses are considered online when the training material is 80% delivered online [34]. Online learning has certain issues as it creates remoteness and is further hampered by a lack of electronic resources and inconsistency in Internet connection [61].

#### 2.5.3. Offline Learning

Offline study is also referred to as self-study, through which students obtain basic knowledge to utilize during online and face-to-face sessions [51]. According to Barthimeus [61], the self-study modules, printed materials, and learning media from things in the surrounding environment are all examples of offline learning. For offline learning, teachers give students the modules, notes, and assigned tasks for offline learning [61]. Moreover, [65] highlighted the online–offline and blended learning methods, witnessing a paradigm shift with rapid and ongoing technological advancement.

In-service training of vocational teachers was conducted by adopting various modes of learning (i.e., face-to-face, online, and offline learning). Lectures, presentations, group activities, and multimedia were used for classroom learning or face-to-face learning. Computers, laptops, Internet, WhatsApp, cell phone, and Gmail were used for online learning while training manuals in the form of CD ROM, notes, and a digital library were used for offline learning. Graham, Woodfield, and Harrison [21], Dawson and Dana [66], and Lei and So [67] believed that the future of the educational approach was a blended learning approach that combined the benefits of face-to-face and online learning. Previous research also highlighted that the success of the professional development of teachers with a blended learning approach depends on the good planning and organization of the training program [34]. According to Nortvig, Petersen, and Balle [32], blended learning should cover at least a 50% training part in face-to-face teaching [68]. We developed a conceptual framework for this study by keeping in mind the above discussion (see Figure 1).

## 3. Research Methodology

A pragmatic paradigm was used for the present research. The aforementioned paradigm supports the use of mixed methods research through a research questionnaire and semi-structured interviews. Tashakkori and Teddlie [68] are of the view that pragmatism is a repudiation of choice between constructivism and positivism while using the epistemology and approaches rather than implementing both viewpoints. The pragmatism approach believes in what works well instead of captivating an extreme position such as that taken by post-positivism or interpretivism. Therefore, pragmatism is a mixed methods research approach that encompasses both quantitative and qualitative assumptions. It is up to the researchers to identify which method, technique, or procedure best suits them as per their need and purposes [69]. Now, the conflict is over among the researchers regarding posing their choice of approach as superior over others, as has happened in the past [68,69].

Quantitative and qualitative data are essential to have a comprehensive understanding about a research problem in spite of using a single approach alone [69]. Mixed methods research should be applied to explore the research question in a better way, particularly when the focus is to collect data from different sources on the same topic [70].

The interviews were designed based on the research objectives and questions. A survey was undertaken by the training participants. The semi-structured interviews were conducted from the Principals and Admission and Placement Officers (APOs) to endorse the results of a survey conducted by the training participants. Both the quantitative and qualitative methods were applied simultaneously in this research study. The reason behind using both methods was to analyze the same phenomenon from different angles. According to Baxter and Jack [71], the topic can be explored through different lenses by applying different sources.

This study used an explanatory sequential mixed methods approach. First, we collected quantitative data through a questionnaire, and then we analyzed the data. This was followed by quantitative data analysis, where we explained the findings of quantitative data through qualitative data using semi-structured interviews. It is a type of design in which one dataset supports the other dataset that was obtained through different research approaches such as qualitative and quantitative research approaches. According to Creswell and Creswell [69], it is not possible in every situation to draw conclusions merely based on one method as both approaches support each other. The quantitative data are dominant in this design. The results of the quantitative analysis will be endorsed by qualitative results.

### 3.1. Quantitative Research

Researchers provide a consultancy to the Punjab Vocational Training Council to conduct need analysis surveys for the training of in-service TVET teachers, measure the effectiveness of the training program, and develop training programs based on research. This study consisted of a population of all of the trained in-service TVET teachers by GIZ under the TVET Reform Program at the institutes of Punjab Vocational Training Council (PVTC). The total number of trained teachers was 781 and these were distributed in 204 vocational training institutes across Punjab affiliated with the PVTC. A census method was used to survey all of the training participants. The census method is used when the population size is small, thus it involves the whole population to obtain intensive information [72]. We sent 803 questionnaires to the participants, and in response, we received 781 useable questionnaires with a response rate of 97.26%. We removed questionnaires from the data that had incomplete information. Therefore, in order to obtain intensive information on all, the trained teachers were asked to fill in the training evaluation questionnaire. This was undertaken to attain more accurate and reliable results.

#### 3.1.1. Instruments of the Study

The training Attitude Questionnaire (TAQ) developed by Homklin [73] was adopted and used with permission in the present study for the collection of quantitative data. It was translated into “Urdu” by an expert in order to facilitate the respondents of the sample of the study. The questionnaire was discussed with a language expert for translation. Certain changes in light of discussion with the supervisor were made so that a clear understanding of the questionnaire was easy for the respondents. Keeping in mind the objectives, research questions, literature [21,67,74], and conceptual framework of the current study, the survey questions were converted to the three-dimensional response (i.e., face-to-face, online, and offline) as the training under study followed the blended learning approach. The agreed questionnaire was then sent to the experts for its validation on the scale of highly satisfied, satisfied, and dissatisfied. The questionnaires were sent through emails and hard forms. Certain changes in the demographic portion and questions were highlighted by the experts. The questionnaire was then validated by the expert by adding these changes. The whole process took 1 to 2 months to obtain a questionnaire validated by the experts. Pilot testing was run after validation of the research questionnaires from four experts. This questionnaire had two parts.

The first part was the demographics: age, experience, gender, occupational trade, and location of the training institutions were added as demographics in this questionnaire.

The second part was on the attitude of the training participants toward different learning approaches. TAQ used a 5-point Likert type scale (i.e., 1 = strongly disagree, 2 = disagree, 3 = neutral, 4 = agree and 5 = strongly agree). The questionnaire was divided into three constructs: first was the in-service vocational teachers’ attitude toward the online learning approach; the second construct was about the in-service vocational teachers’ attitude toward the face-to-face learning approach; and the third construct was about the in-service vocational teachers’ attitude toward the offline learning approach.

#### 3.1.2. Pilot Testing of the Instrument

Pilot testing was run after the validation of the research questionnaires from four experts. Permission from the competent authority of PVTC was also sought out. The researchers signed a non-disclosure agreement with the HR department before the data collection in the pilot phase. A Google docs file was sent to the respondents to fill out the questionnaires.

The data received in the form of Google docs was converted into an Excel file and then the data were imported in SPSS version 22. The CSV file was produced to upload the data file in Smart_PLS software.

All items showed a factor loading greater than 0.4 with their relevant factors, as recommended by researchers (see Figure 2). The reliability and validity of the questionnaire was measured with the help of Cronbach’s alpha, rho alpha, and composite reliability. It was found that online, offline, and face-to-face constructs showed a Cronbach’s alpha, rho_Alpha, and composite reliability values higher than 0.7, as recommended by Hair Jr. et al. [75]. The average variance extracted (AVE) values for three constructs were found to be higher than the threshold of 0.5, as recommended by Hair Jr., Sarstedt, Ringle, and Gudergan [75] (See Table 1).

HTMT is a new technique to measure the discriminant validity of the constructs. All three constructs (i.e., face-to-face, online, and offline learning approaches) showed HTMT values less than 0.8, as recommended by the researchers (as given in Table 2).

The ANOVA was applied on the various modes of training (i.e., face-to-face, online, and offline) to compare the attitudes of the training participants among them. Table 3 clearly shows a significant value of less than 0.05 (*p* < 0.05), indicating that the participants had different levels of attitude for different modes of learning. In order to check the difference of opinion among the participants, the post hoc test was applied, as shown in Table 4. The LSD test for attitude toward the training showed that respondents were more satisfied with face-to-face interaction than the online phase (MD = −0.202, SD = 0.031, *p* < 0.05). Similarly, the respondents were more satisfied by online learning in comparison to the offline learning phase (MD = −0.166, SD = 0.031, *p* < 0.05). Thus, the face-to-face learning mode was preferred the most over the online and offline learning modes.

### 3.2. Qualitative Research

There are many designs of qualitative research available such as ethnography, grounded theory, phenomenology, narrative analysis, and case study. The present study used a multiple case study design. Different respondents such as principals, teachers, admission and placement officers (APOs) of the vocational training institutes were included in this study to explore the appropriate combination of the blended learning approaches. A single case cannot provide a broad perspective regarding the research, only multiple cases can do this (Theiler, 2012). Therefore, more accurate and convincing data can be obtained through multiple cases [76]. We analyzed the semi-structured interviews with the principals, admission and placement officers, and teachers.

The purposive sampling technique was used to conduct semi-structured interviews with the training participants (*n* = 15), admission and placement officers (*n* = 15), and principals (*n* = 15) of the vocational training institutes until the interviews reached the saturation point. Although we obtained saturations for all three categories before reaching the fifteenth interview for each. However, we conducted 15 interviews in each category to ensure the maximum level of saturation. All of the training participants, teachers, admission and placement officers, and principals were purposefully selected, who served as teachers and then had been promoted to their current designation. They also attended the in-service vocational teacher training program through the blended learning approach. This was conducted only because of the fact that accurate data can be collected through the trained individuals who not only implemented the training in their classrooms, but also had the experience of the whole institute from the different lens of the principals and admission and placement officers.

Interview protocols were established that covered the research objectives and research questions. Interview protocols were shared with the five experts and any amendments were made in the protocols.

Qualitative data analysis is a composite process as described by Goodwin and Goodwin [77] because a large amount of data are collected and arranged in the form of themes and categories to create theories. In order to convert all codes into themes, reading and re-reading of ideas, words, and phrases in the segment should be the focus [69].

Interviews were conducted with the respondents and qualitative data were obtained, which is a primary source of data for the present study. Qualitative thematic analysis was conducted of the data obtained through semi-structured interviews [78]. Interviews were conducted in Urdu and then translated into English by listening several times. The transcriptions were reviewed by an English language expert and appropriate grammatical changes were made [79]. A summary of the responses of the participants was made through thematic analysis under three themes (i.e., online, face-to-face, and offline).

#### 3.2.1. Face-to-Face Phase

The respondents had different point of views regarding the duration of the training. One of the respondents reported, “I add one thing, that it should be maximum time for face-to-face learning”. He further explained that “if you allocate 70% time for the face-to-face mode, then you can discuss new ideas and problems faced by you with the available trainer to resolve mutually. Because DVD and online data that was provided was such that you can do it properly only when you are in IT related field Otherwise it is difficult, if you are from non-IT related field” (Teacher Participant 1). Another respondent said, “Duration of the training was good and one-week time in a month for face-to-face was also good”. He further elaborated his point that his training period comprised of three months out of which one week was dedicated for face-to-face mode in each month. He was of the view that if you undertake half an hour daily for offline and online work, then you can prepare your tasks and assignments easily because you have to perform your office work as well along with the assigned tasks during face-to-face training. He said that it was an ideal combination of face-to-face, online, and offline learning” (Teacher Participant 2). From the viewpoint of another respondent: “Thirty % time should be for face-to-face mode that promotes learning by discussion in classroom” (Teacher Participant 3). One of the respondents said: “Forty % time should be given to the face-to-face training mode”. He further explained that “this is because you learn the use of technology, Internet, communication, conversation, use of printer, scanner during training” (Teacher Participant 4). Another respondent said that “Face-to-face training mode was much better and easy to understand so 80% training should be face-to-face” (Teacher Participant 5).

All of the respondents were clear about the importance of face-to-face learning, but had different views about the duration of the phase. One of the respondents reported: “*I think duration of the training is enough but split of the training should be such that face-to-face 70 to 75%,” (APO Participant 1)*. Other respondent said that: “*Face-to-face training was very good as the trainer delivered lecturers were very knowledgeable. I strongly say that it should be 90% (APO Participant 2).* Another respondent was of the view that: “*Most important learning dimension is face-to-face and it must be 80%” (Principal Participant 4).* Another respondent said: “*Face-to-face phase should be 60%” (Teacher Participant 5).* The last respondent said “*Face-to-face duration should be 40%” (Principal Participant 3).*

#### 3.2.2. Online Phase

One of the respondents reported “*Online phase should be 15% of the total time” (Participant 1)*. Another respondent was of the view that: “Three weeks in a month for online mode is good because if you serve half an hour daily for online working then you can prepare your tasks and assignments easily because you have to perform your office work too along with the assigned tasks during face-to-face training” *(Teacher Participant 2)*. According to another respondent, “*Online phase should cover 40% times because in his opinion blended learning is basically an online research that teacher performs himself” (Teacher Participant 3)*. Another respondent was of the view that “*Thirty percent time should be allocated for online mode” (Teacher Participant 4).* According to another respondent: “*there is always something missing in understanding so 5 percent time should be given to online for best working” (Teacher Participant 5)*.

A few respondents were of the view that the online phase should have 20% weighting while the other said that: “It should be 40%” *(APO Participant 3)*. One of the respondents said that: “*It should be 10%” (Principal Participant 4)* and the other argued that: “It should be 5 percent” (APO Participant 2)

#### 3.2.3. Offline Phase

When we analyzed all of the training modes, we can come to the conclusion that many respondents are of the view that face-to-face is better in understanding concepts and resolving issues so the face-to-face mode should be given more time compared to the online and offline phases because the trainers also had to perform their official working activities during the offline phase. A few respondents were also of the view that a similar time should be allocated for online and offline working. If we take an average of all respondents, then 55, 22.5, and 22.5% time should be allocated for face-to-face, online, and offline modes, respectively. One of the respondents reported: “*Three weeks in a month for offline mode is good because*
*if you serve half an hour daily for offline assignments then you can prepare your assignments easily because you have to perform your office work too along with the assigned tasks during face-to-face training*
*(Teacher Participant 2).* From another respondent: “*Thirty percent time should be allocated for offline mode*” *(Teacher Participant 3)*. Another respondent was of the view that: “*Thirty percent time should be allocated for offline mode*
*(Teacher Participant 4)*. According to another respondent: “*Offline working also create problems because you have to perform your working activities. You find very less time for offline working so 5 percent time should be for offline assignments*” *(Teacher Participant 5)*.

A few of the respondents’ emphasized that the offline duration should be 20% while a few other respondents argued that it should be 10%. One of the respondents said that: “*It should be 5 percent” (Principal Participant 2)*.

When we look at all training modes, then we can to the conclusion that all respondents were of the view that face-to-face was better, so the face-to-face mode should be given more time compared to the online and offline phases. If we take an average of all respondents, then 68, 19, and 13 percent time should be allocated for the face-to-face, online, and offline modes, respectively.

## 4. Discussion

The NAVTTC and GIZ applied blended learning approaches for in-service vocational teacher training programs in Pakistan. It was the aim of the study to find the appropriate blend of the online, face-to-face, and offline approaches for in-service vocational teacher training in Pakistan. According to our best knowledge, this study is among the pioneer research to find the appropriate blend of in-service vocational teachers in the context of a developing country such as Pakistan.

We operationalized the blended learning approach in our study as a combination of three modes: online, offline, and face-to-face. The most difficult type of learning in terms of challenges was online, followed by distance learning, and finally, blended learning [57]. Online learning provides solutions to the problems faced in conventional education; however, it also imposes some problems (e.g., separation, remoteness, adequate feedback, distancing from group members, and lack of seriousness) [62]. A previous study has shown that 89% students believed that practical training through an online mode of delivery was not recommended [63]. In this critical situation, blended learning has emerged as a solution to the problems faced in both approaches as it covers online and face-to-face learning [64]. Koi-Akrofi, Owusu-Oware, and Tanye [57] argued that blended learning combines the use of technology with traditional classroom activities to promote learner confidence and competence. According to Chen and Jones [80], further advantages of blended learning include a comprehensive understanding of the subjects through the use of web-based resources and the active participation of students in the classroom. Teachers and students were more satisfied with the blended learning approach compared to the single approach due to its effectiveness. It provides the facility by using different resources that improve the interaction between students and teachers and self-paced learning [62]. The previous research showed that 83% students believed that a blended learning approach had better results and so was their first choice [63]. The results of the quantitative data revealed that face-to-face learning is much more preferable compared to online and offline learning. The results of the previous research have endorsed the results of the current study that the student’s preference for face-to-face learning is higher than that of online and offline learning [81,82,83]. According to Baker and Unni [52], the preference of students from Asian regions for face-to-face learning was higher compared to online and offline learning. Participants felt comfortable in the face-to-face session as it provided a quick solution to their problems and hands-on experience. The authors in [53] argued that student satisfaction with face-to-face learning was greater than in online learning. In contrast to the current results, research has also shown that students were more satisfied from the online learning compared to face-to-face learning. Second, online learning is preferred as it provides virtual contact with the fellows and trainer in which ideas, problems, and their solutions can be discussed with the independence of time and place [84]. Offline learning was the least preferred option in comparison to the face-to-face and online modes. Findings of the qualitative interviews of the teachers, principals, and APOs endorsed the results of the quantitative survey and revealed that face-to-face learning was preferred the most compared to online and offline learning. It further suggests that 61.5% time should be allocated for face-to-face learning, whereas 20.75 and 17.75% time should be allocated for online and offline learning, respectively, for better learning by applying a blended learning approach for in-service vocational education teacher training. Graham, Woodfield, and Harrison [21], Dawson [74], and Lei and So [67] believe that the future of the educational approach is a blended learning approach that combines the benefits of face-to-face and online learning. Previous researches have also highlighted that the success of the professional development of teachers with a blended learning approach depends on the good planning and organization of the training program [34]. According to Nortvig, Petersen, and Balle [32], blended learning should cover at least a 50% training part in face-to-face teaching [32]. It means that the remaining 50% time should be allocated for the online and offline modes. According to a case study that supports the findings of the present research states that a face-to-face program is a better fit because of the increased in-class support, higher social presence, and improved interaction with the instructor [85]. Lu and Lemonde [86] discussed that face-to-face training increased higher-order learning better than online education. Horn and Staker [36] argued that while talking about blended learning, one should refrain from time bounding during online and face-to-face educational practices. When the training material was delivered 80% online, then the course was considered as online [34].

## 5. Conclusions

The present research clearly identified that face-to-face learning was preferred and demanded compared to online and offline learning. Findings of the qualitative interviews of the principals and APOs endorsed the results of the quantitative survey. The study finally extracted that 61.5%, 20.75%, and 17.75% of time should be allocated for face-to-face, online, and offline learning, respectively. It was also suggested to only allocate two tasks to the training participants for the independent learning phase of the training.

The face-to-face phase of training should be given more time than the online and offline phases because of the nature of vocational skills, which is practical based and needs hands on experience. For better results, 61.5% time should be allocated for face-to-face learning whereas 20.75 and 17.75% of time should be allocated for online and offline learning, respectively. The training providers should adopt the above format to plan training with blended learning approaches in the future to gain the maximum benefits from the innovative approach. Vocational training institutes can benefit from this research. VTIs should utilize all sorts of learning approaches for the continuous professional development of in-service vocational teachers. Teacher training institutes must conduct a needs analysis to understand the training requirements of the in-service teachers. They can then develop a training program based on the need analysis. These training programs can be launched through face-to-face, online, and offline learning approaches. In-service teachers and teacher trainers can be invited to central training centers to deliver the greatest part of the training program. Then, the in-service teachers can be sent home to perform the tasks offline while sharing the training assignments online with their colleagues and master trainers. The training cycle of face-to-face, online, and offline learning can be repeated by keeping in mind the length and training requirements. It is necessary that in-service teachers should be trained mostly through face-to-face than online, while offline should be the minimum component of the training program.

## 6. Future Research

The in-service vocational education teacher training with blended learning approaches was also conducted in other provinces (i.e., Khyber Pakhtoon Khawa, Sindh, and Baluchistan), so its replica can be conducted in those provinces. Research can also be conducted to compare the effectiveness of blended learning between in-service teachers in the vocational and technical streams.

## Figures and Tables

**Figure 1 ijerph-19-10668-f001:**
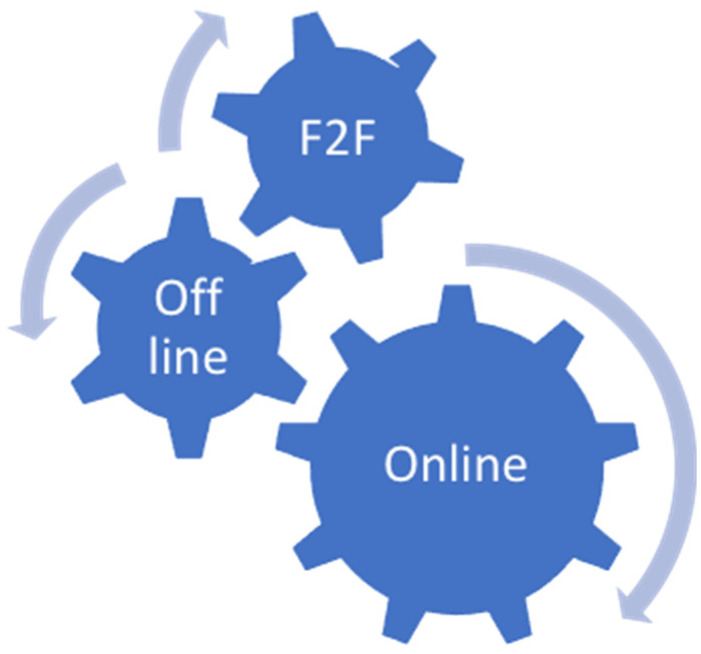
The conceptual framework.

**Figure 2 ijerph-19-10668-f002:**
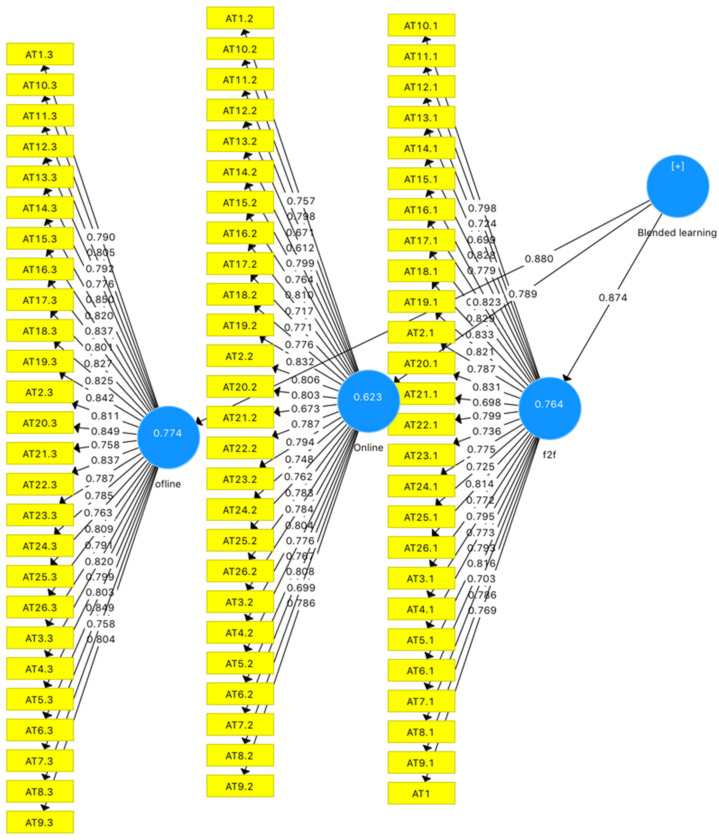
The factor loading of the items.

**Table 1 ijerph-19-10668-t001:** The reliability of the constructs.

	Cronbach	rho_A	CR	(AVE)
Online	0.972	0.972	0.974	0.588
f2f	0.974	0.975	0.976	0.609
offline	0.979	0.979	0.98	0.652

**Table 2 ijerph-19-10668-t002:** The discriminant validity (HTMT).

	Online	f2f	Offline
Online			
f2f	0.55		
offline	0.54	0.689	

**Table 3 ijerph-19-10668-t003:** ANOVA for various modes of training (i.e., face-to-face, online, and offline).

		Mean Square	F	Sig.
AT	Between Groups	16.297	52.216	0.000
	Within Groups	0.312		
	Total			
	Total			

AT = Attitude toward different training modes.

**Table 4 ijerph-19-10668-t004:** Post hoc for various modes of training (i.e., face-to-face, online, and offline).

Dependent Variable	(I) Groups	(J) Groups	Mean Difference (I–J)	Std. Error	Sig.
AT	online	F2F	−0.20202 *	0.03150	0.000
		offline	0.11606 *	0.03150	0.000
	F2F	online	0.20202 *	0.03150	0.000
		offline	0.31807 *	0.03150	0.000
	offline	online	−0.11606 *	0.03150	0.000
		F2F	−0.31807 *	0.03150	0.000

AT = Attitude toward different training modes. * = Significant of Mean Difference (I–J).

## Data Availability

The data will be provided at time of the request.

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
