# Peer review of "Analyzing an Appropriate Blend of Face-to-Face, Offline and Online Learning Approaches for the In-Service Vocational Teacher’s Training Program"

_ijerph, 2022, doi:10.3390/ijerph191710668_

Round 1

Reviewer 1 Report

Dear Authors,

Here are my comments:

1-The tittle focus on adapting a blended learning design. But, the research is abouth comparing 3 types of blended types. So, the tittle need to be re-considered.

2-The related literature can be updated with new studies published in last 2 years.

3-Research questions are not clear. Need to be clearified. Additionaly, I didnot understand th H0 of "Blended learning approaches needs specific weightage of online, face-to-face and 331 offline approaches for in-service vocation teacher’s training program ". Maybe better to remove.

4-The design of mixed method is not clear. Research design is a explanatory mixed method design (first quant. than qualt. to explain first findings). Should be considered.

5-This research is a study that should offer practical suggestions due to its subject. Suggestions for practice can be provided.

Best

Author Response

Comment 1-The tittle focus on adapting a blended learning design. But, the research is abouth comparing 3 types of blended types. So, the tittle need to be re-considered.

Answer 1: We have amend the Title according to the feedback of the reviewer. Kindly check with tracked changes.

Comment 2-The related literature can be updated with new studies published in last 2 years.

Answer 2: We have added a few latest studies in the literature and introduction according to your valuable feedback. Kindly check with tracked changes.

Comment 3-Research questions are not clear. Need to be clearified. Additionaly, I didnot understand th H0 of "Blended learning approaches needs specific weightage of online, face-to-face and 331 offline approaches for in-service vocation teacher’s training program ". Maybe better to remove.

Answer 3: We have revised research questions and removed H0 as per your feedback. 

Comment 4-The design of mixed method is not clear. Research design is a explanatory mixed method design (first quant. than qualt. to explain first findings). Should be considered.

Answer 4: we have made changes according to your feedback in the methodological sections and explained explanatory mixed method design.

Comment 5-This research is a study that should offer practical suggestions due to its subject. Suggestions for practice can be provided.

Answer 5: We have provided some practical suggestions. Kindly check with tracked Changes. 

Regards

Reviewer 2 Report

Your study has merit. I see four things which should be addressed. 

1. Explain how your conceptual model and liter ids informed selection of your instrument and development of your interview guide. 

2. indicate the response rate on the quantitative portion and if necessary how you handled no response error. 

3. Explain your selection process for interviews in more detail. You say that you interviewed until saturation as met, but you had 15 in each category. Seems unlikely that all 3 groups would reach saturation with exactly the same number. 

4. Explain the researchers’ connection to the participants. 

Author Response

Reviewer 2:

Comment 1. Explain how your conceptual model and liter ids informed selection of your instrument and development of your interview guide. 

Answer 1: We have mentioned it now in : section 3.1.1. Instruments of the Study

Training Attitude Questionnaire (TAQ) developed by 73 was adopted and used with the permission in the present study for the collection of quantitative data. It was translated in "Urdu" by an expert in order to facilitate the respondents of the sample of the study. The questionnaire was discussed with the language expert for translation. Certain changes in the light of discussion with the supervisor were made so that a clear understanding of the questionnaire may become easy for the respondents. Keeping in view the objectives, research questions, literature 21, 67, 74 and conceptual framework of the current study survey questions were converted to the three-dimensional response i.e., face to face, online and offline as the training under study followed blended learning approach. The agreed questionnaire was then sent to the experts for its validation on the scale of highly satisfied, satisfied and dissatisfied. The questionnaires were sent through emails and hard forms. Certain changes were in the demographic portion and questions were highlighted by the experts.

Comment 2. indicate the response rate on the quantitative portion and if necessary how you handled no response error. 

Answer 2: Thank you for highlighting the issue. We have mentioned now. We sent 803 questionnaires to the participants, in response, we received 781 useable questionnaires with response rate of 97.26%. We removed the questionnaire from the data which had incomplete information.

Comment 3. Explain your selection process for interviews in more detail. You say that you interviewed until saturation as met, but you had 15 in each category. Seems unlikely that all 3 groups would reach saturation with exactly the same number. 

Answer 3: Although some categories have shown saturation level even before 15 interviews. But we conducted 15 interviews for each category to make sure the saturation.

Comment 4. Explain the researchers’ connection to the participants. 

Answer 4: Researchers are working as consultant for Punjab Vocational Training Council. We provide them consultancy to conduct needs analysis surveys for their teacher training program, assessment of the effectiveness of in-service teacher’s training and etc.